# End-to-End Kernel Learning with Supervised Convolutional Kernel Networks

**Julien Mairal**
Inria*
julien.mairal@inria.fr

## Abstract

In this paper, we introduce a new image representation based on a multilayer kernel machine. Unlike traditional kernel methods where data representation is decoupled from the prediction task, we learn how to shape the kernel with supervision. We proceed by first proposing improvements of the recently-introduced convolutional kernel networks (CKNs) in the context of unsupervised learning; then, we derive backpropagation rules to take advantage of labeled training data. The resulting model is a new type of convolutional neural network, where optimizing the filters at each layer is equivalent to learning a linear subspace in a reproducing kernel Hilbert space (RKHS). We show that our method achieves reasonably competitive performance for image classification on some standard "deep learning" datasets such as CIFAR-10 and SVHN, and also for image super-resolution, demonstrating the applicability of our approach to a large variety of image-related tasks.

## 1  Introduction

In the past years, deep neural networks such as convolutional or recurrent ones have become highly popular for solving various prediction problems, notably in computer vision and natural language processing. Conceptually close to approaches that were developed several decades ago (see, [13]), they greatly benefit from the large amounts of labeled data that have been made available recently, allowing to learn huge numbers of model parameters without worrying too much about overfitting. Among other reasons explaining their success, the engineering effort of the deep learning community and various methodological improvements have made it possible to learn in a day on a GPU complex models that would have required weeks of computations on a traditional CPU (see, *e.g.*, [10, 12, 23]).

Before the resurgence of neural networks, non-parametric models based on positive definite kernels were one of the most dominant topics in machine learning [22]. These approaches are still widely used today because of several attractive features. Kernel methods are indeed versatile; as long as a positive definite kernel is specified for the type of data considered—*e.g.*, vectors, sequences, graphs, or sets—a large class of machine learning algorithms originally defined for linear models may be used. This family include supervised formulations such as support vector machines and unsupervised ones such as principal or canonical component analysis, or K-means and spectral clustering. *The problem of data representation is thus decoupled from that of learning theory and algorithms.* Kernel methods also admit natural mechanisms to control the learning capacity and reduce overfitting [22].

On the other hand, traditional kernel methods suffer from several drawbacks. The first one is their computational complexity, which grows quadratically with the sample size due to the computation of the Gram matrix. Fortunately, significant progress has been achieved to solve the scalability issue, either by exploiting low-rank approximations of the kernel matrix [28, 31], or with random sampling techniques for shift-invariant kernels [21]. The second disadvantage is more critical; by decoupling

learning and data representation, kernel methods seem by nature incompatible with end-to-end learning—that is, the representation of data adapted to the task at hand, which is the cornerstone of deep neural networks and one of the main reason of their success. *The main objective of this paper is precisely to tackle this issue in the context of image modeling.*

Specifically, our approach is based on convolutional kernel networks, which have been recently introduced in [18]. Similar to hierarchical kernel descriptors [3], local image neighborhoods are mapped to points in a reproducing kernel Hilbert space via the kernel trick. Then, hierarchical representations are built via kernel compositions, producing a sequence of "feature maps" akin to convolutional neural networks, but of infinite dimension. To make the image model computationally tractable, convolutional kernel networks provide an approximation scheme that can be interpreted as a particular type of convolutional neural network learned without supervision.

To perform end-to-end learning given labeled data, *we use a simple but effective principle consisting of learning discriminative subspaces in RKHSs, where we project data.* We implement this idea in the context of convolutional kernel networks, where linear subspaces, one per layer, are jointly optimized by minimizing a supervised loss function. The formulation turns out to be a new type of convolutional neural network with a non-standard parametrization. The network also admits simple principles to learn without supervision: learning the subspaces may be indeed achieved efficiently with classical kernel approximation techniques [28, 31].

To demonstrate the effectiveness of our approach in various contexts, we consider image classification benchmarks such as CIFAR-10 [12] and SVHN [19], which are often used to evaluate deep neural networks; then, we adapt our model to perform image super-resolution, which is a challenging inverse problem. On the SVHN and CIFAR-10 datasets, we obtain a competitive accuracy, with about $2\%$ and $10\%$ error rates, respectively, without model averaging or data augmentation. For image up-scaling, we outperform recent approaches based on classical convolutional neural networks [7, 8].

We believe that these results are highly promising. Our image model achieves competitive performance in two different contexts, paving the way to many other applications. Moreover, our results are also subject to improvements. In particular, we did not use GPUs yet, which has limited our ability to exhaustively explore model hyper-parameters and evaluate the accuracy of large networks. We also did not investigate classical regularization/optimization techniques such as Dropout [12], batch normalization [11], or recent advances allowing to train very deep networks [10, 23]. To gain more scalability and start exploring these directions, we are currently working on a GPU implementation, which we plan to publicly release along with our current CPU implementation.

**Related Deep and Shallow Kernel Machines.**   One of our goals is to make a bridge between kernel methods and deep networks, and ideally reach the best of both worlds. Given the potentially attractive features of such a combination, several attempts have been made in the past to unify these two schools of thought. A first proof of concept was introduced in [5] with the arc-cosine kernel, which admits an integral representation that can be interpreted as a one-layer neural network with random weights and infinite number of rectified linear units. Besides, a multilayer kernel may be obtained by kernel compositions [5]. Then, hierarchical kernel descriptors [3] and convolutional kernel networks [18] extend a similar idea in the context of images leading to unsupervised representations [18].

Multiple kernel learning  [24] is also related to our work since is it is a notable attempt to introduce supervision in the kernel design. It provides techniques to select a combination of kernels from a predefined collection, and typically requires to have already "good" kernels in the collection to perform well. More related to our work, the backpropagation algorithm for the Fisher kernel introduced in [25] learns the parameters of a Gaussian mixture model with supervision. In comparison, our approach does not require a probabilistic model and learns parameters at several layers. Finally, we note that a concurrent effort to ours is conducted in the Bayesian community with deep Gaussian processes [6], complementing the Frequentist approach that we follow in our paper.

## 2   Learning Hierarchies of Subspaces with Convolutional Kernel Networks

In this section, we present the principles of convolutional kernel networks and a few generalizations and improvements of the original approach of [18]. Essentially, the model builds upon four ideas that are detailed below and that are illustrated in Figure 1 for a model with a single layer.

**Idea 1: use the kernel trick to represent local image neighborhoods in a RKHS.**
Given a set $\mathcal{X}$, a positive definite kernel $K : \mathcal{X} \times \mathcal{X} \to \mathbb{R}$ implicitly defines a Hilbert space $\mathcal{H}$, called reproducing kernel Hilbert space (RKHS), along with a mapping $\varphi : \mathcal{X} \to \mathcal{H}$. This embedding is such that the kernel value $K(\mathbf{x}, \mathbf{x}')$ corresponds to the inner product $\langle \varphi(\mathbf{x}), \varphi(\mathbf{x}') \rangle_{\mathcal{H}}$. Called "kernel trick", this approach can be used to obtain nonlinear representations of local image patches [3, 18].

More precisely, consider an image $I_0 : \Omega_0 \to \mathbb{R}^{p_0}$, where $p_0$ is the number of channels, *e.g.*, $p_0 = 3$ for RGB, and $\Omega_0 \subset [0, 1]^2$ is a set of pixel coordinates, typically a two-dimensional grid. Given two image patches $\mathbf{x}, \mathbf{x}'$ of size $e_0 \times e_0$, represented as vectors in $\mathbb{R}^{p_0 e_0^2}$, we define a kernel $K_1$ as

$$K_1(\mathbf{x}, \mathbf{x}') = \|\mathbf{x}\| \, \|\mathbf{x}'\| \, \kappa_1\left(\left\langle \frac{\mathbf{x}}{\|\mathbf{x}\|}, \frac{\mathbf{x}'}{\|\mathbf{x}'\|} \right\rangle\right) \text{ if } \mathbf{x}, \mathbf{x}' \neq 0 \text{ and } 0 \text{ otherwise,} \qquad (1)$$

where $\|.\|$ and $\langle ., . \rangle$ denote the usual Euclidean norm and inner-product, respectively, and $\kappa_1(\langle ., . \rangle)$ is a dot-product kernel on the sphere. Specifically, $\kappa_1$ should be smooth and its Taylor expansion have non-negative coefficients to ensure positive definiteness [22]. For example, the arc-cosine [5] or the Gaussian (RBF) kernels may be used: given two vectors $\mathbf{y}, \mathbf{y}'$ with unit $\ell_2$-norm, choose for instance

$$\kappa_1(\langle \mathbf{y}, \mathbf{y}' \rangle) = e^{\alpha_1(\langle \mathbf{y}, \mathbf{y}' \rangle - 1)} = e^{-\frac{\alpha_1}{2}\|\mathbf{y} - \mathbf{y}'\|_2^2}. \qquad (2)$$

Then, we have implicitly defined the RKHS $\mathcal{H}_1$ associated to $K_1$ and a mapping $\varphi_1 : \mathbb{R}^{p_0 e_0^2} \to \mathcal{H}_1$.

**Idea 2: project onto a finite-dimensional subspace of the RKHS with convolution layers.**
The representation of patches in a RKHS requires finite-dimensional approximations to be computationally manageable. The original model of [18] does that by exploiting an integral form of the RBF kernel. Specifically, given two patches $\mathbf{x}$ and $\mathbf{x}'$, convolutional kernel networks provide two vectors $\psi_1(\mathbf{x}), \psi_1(\mathbf{x}')$ in $\mathbb{R}^{p_1}$ such that the kernel value $\langle \varphi_1(\mathbf{x}), \varphi_1(\mathbf{x}') \rangle_{\mathcal{H}_1}$ is close to the Euclidean inner product $\langle \psi_1(\mathbf{x}), \psi_1(\mathbf{x}') \rangle$. After applying this transformation to all overlapping patches of the input image $I_0$, a spatial map $M_1 : \Omega_0 \to \mathbb{R}^{p_1}$ may be obtained such that for all $z$ in $\Omega_0$, $M_1(z) = \psi_1(\mathbf{x}_z)$, where $\mathbf{x}_z$ is the $e_0 \times e_0$ patch from $I_0$ centered at pixel location $z$.[2] With the approximation scheme of [18], $M_1$ can be interpreted as the output feature map of a one-layer convolutional neural network.

A conceptual drawback of [18] is that data points $\varphi_1(\mathbf{x}_1), \varphi_1(\mathbf{x}_2), \ldots$ are approximated by vectors that do not live in the RKHS $\mathcal{H}_1$. This issue can be solved by using variants of the Nyström method [28], which consists of projecting data onto a subspace of $\mathcal{H}_1$ with finite dimension $p_1$. For this task, we have adapted the approach of [31]: we build a database of $n$ patches $\mathbf{x}_1, \ldots, \mathbf{x}_n$ randomly extracted from various images and normalized to have unit $\ell_2$-norm, and perform a spherical $K$-means algorithm to obtain $p_1$ centroids $\mathbf{z}_1, \ldots, \mathbf{z}_{p_1}$ with unit $\ell_2$-norm. Then, a new patch $\mathbf{x}$ is approximated by its projection onto the $p_1$-dimensional subspace $\mathcal{F}_1 = \text{Span}(\varphi(\mathbf{z}_1), \ldots, \varphi(\mathbf{z}_{p_1}))$.

The projection of $\varphi_1(\mathbf{x})$ onto $\mathcal{F}_1$ admits a natural parametrization $\psi_1(\mathbf{x})$ in $\mathbb{R}^{p_1}$. The explicit formula is classical (see [28, 31] and Appendix A), leading to

$$\psi_1(\mathbf{x}) := \|\mathbf{x}\| \kappa_1(\mathbf{Z}^\top \mathbf{Z})^{-1/2} \kappa_1\left(\mathbf{Z}^\top \frac{\mathbf{x}}{\|\mathbf{x}\|}\right) \text{ if } \mathbf{x} \neq 0 \text{ and } 0 \text{ otherwise,} \qquad (3)$$

where we have introduced the matrix $\mathbf{Z} = [\mathbf{z}_1, \ldots, \mathbf{z}_{p_1}]$, and, by an abuse of notation, the function $\kappa_1$ is applied pointwise to its arguments. Then, the spatial map $M_1 : \Omega_0 \to \mathbb{R}^{p_1}$ introduced above can be obtained by (i) computing the quantities $\mathbf{Z}^\top \mathbf{x}$ for all patches $\mathbf{x}$ of the image $I$ (spatial convolution after mirroring the filters $\mathbf{z}_j$); (ii) contrast-normalization involving the norm $\|\mathbf{x}\|$; (iii) applying the pointwise non-linear function $\kappa_1$; (iv) applying the linear transform $\kappa_1(\mathbf{Z}^\top \mathbf{Z})^{-1/2}$ at every pixel location (which may be seen as $1 \times 1$ spatial convolution); (v) multiplying by the norm $\|\mathbf{x}\|$ making $\psi_1$ homogeneous. In other words, we obtain a particular convolutional neural network, with non-standard parametrization. Note that learning requires only performing a K-means algorithm and computing the inverse square-root matrix $\kappa_1(\mathbf{Z}^\top \mathbf{Z})^{-1/2}$; therefore, the training procedure is very fast.

Then, it is worth noting that the encoding function $\psi_1$ with kernel (2) is reminiscent of radial basis function networks (RBFNs) [4], whose hidden layer resembles (3) without the matrix $\kappa_1(\mathbf{Z}^\top \mathbf{Z})^{-1/2}$ and with no normalization. The difference between RBFNs and our model is nevertheless significant. The RKHS mapping, which is absent from RBFNs, is indeed a key to the multilayer construction that will be presented shortly: a network layer takes points from the RKHS's previous layer as input and use the corresponding RKHS inner-product. To the best of our knowledge, there is no similar multilayer and/or convolutional construction in the radial basis function network literature.

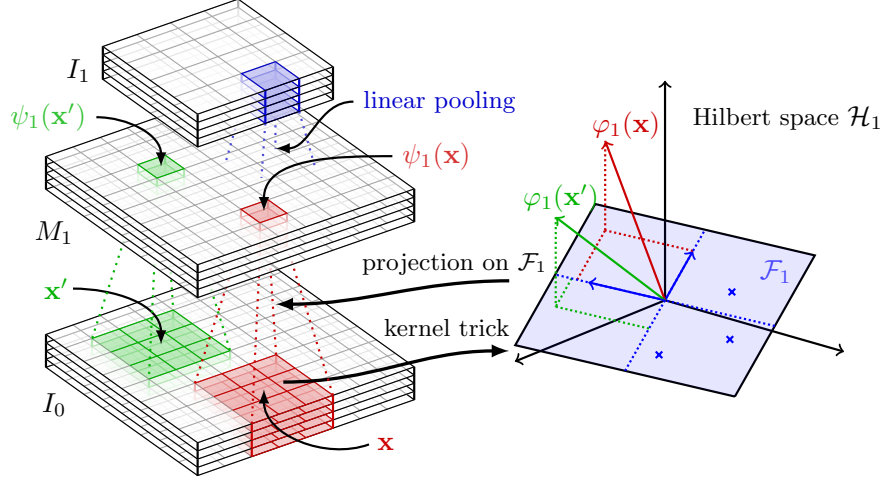

Figure 1: Our variant of convolutional kernel networks, illustrated between layers $0$ and $1$. Local patches (receptive fields) are mapped to the RKHS $\mathcal{H}_1$ via the kernel trick and then projected to the finite-dimensional subspace $\mathcal{F}_1 = \text{Span}(\varphi(\mathbf{z}_1), \ldots, \varphi(\mathbf{z}_{p_1}))$. The small blue crosses on the right represent the points $\varphi(\mathbf{z}_1), \ldots, \varphi(\mathbf{z}_{p_1})$. With no supervision, optimizing $\mathcal{F}_1$ consists of minimizing projection residuals. With supervision, the subspace is optimized via back-propagation. Going from layer $k$ to layer $k + 1$ is achieved by stacking the model described here and shifting indices.

**Idea 3: linear pooling in $\mathcal{F}_1$ is equivalent to linear pooling on the finite-dimensional map $M_1$.**
The previous steps transform an image $I_0 : \Omega_0 \to \mathbb{R}^{p_0}$ into a map $M_1 : \Omega_0 \to \mathbb{R}^{p_1}$, where each vector $M_1(z)$ in $\mathbb{R}^{p_1}$ encodes a point in $\mathcal{F}_1$ representing information of a local image neighborhood centered at location $z$. Then, convolutional kernel networks involve a pooling step to gain invariance to small shifts, leading to another finite-dimensional map $I_1 : \Omega_1 \to \mathbb{R}^{p_1}$ with smaller resolution:

$$I_1(z) = \sum_{z' \in \Omega_0} M_1(z') e^{-\beta_1 \|z' - z\|_2^2}. \tag{4}$$

The Gaussian weights act as an anti-aliasing filter for downsampling the map $M_1$ and $\beta_1$ is set according to the desired subsampling factor (see [18]), which does not need to be integer. Then, every point $I_1(z)$ in $\mathbb{R}^{p_1}$ may be interpreted as a linear combination of points in $\mathcal{F}_1$, which is itself in $\mathcal{F}_1$ since $\mathcal{F}_1$ is a linear subspace. Note that the linear pooling step was originally motivated in [18] as an approximation scheme for a match kernel, but this point of view is not critically important here.

**Idea 4: build a multilayer image representation by stacking and composing kernels.**
By following the first three principles described above, the input image $I_0 : \Omega_0 \to \mathbb{R}^{p_0}$ is transformed into another one $I_1 : \Omega_1 \to \mathbb{R}^{p_1}$. It is then straightforward to apply again the same procedure to obtain another map $I_2 : \Omega_2 \to \mathbb{R}^{p_2}$, then $I_3 : \Omega_3 \to \mathbb{R}^{p_3}$, *etc*. By going up in the hierarchy, the vectors $I_k(z)$ in $\mathbb{R}^{p_k}$ represent larger and larger image neighborhoods (aka. receptive fields) with more invariance gained by the pooling layers, akin to classical convolutional neural networks.

The multilayer scheme produces a sequence of maps $(I_k)_{k \geq 0}$, where each vector $I_k(z)$ encodes a point—say $f_k(z)$—in the linear subspace $\mathcal{F}_k$ of $\mathcal{H}_k$. Thus, we implicitly represent an image at layer $k$ as a spatial map $f_k : \Omega_k \to \mathcal{H}_k$ such that $\langle I_k(z), I'_k(z') \rangle = \langle f_k(z), f'_k(z') \rangle_{\mathcal{H}_k}$ for all $z, z'$. As mentioned previously, the mapping to the RKHS is a key to the multilayer construction. Given $I_k$, larger image neighborhoods are represented by patches of size $e_k \times e_k$ that can be mapped to a point in the Cartesian product space $\mathcal{H}_k^{e_k \times e_k}$ endowed with its natural inner-product; finally, the kernel $K_{k+1}$ defined on these patches can be seen as a kernel on larger image neighborhoods than $K_k$.

## 3 End-to-End Kernel Learning with Supervised CKNs

In the previous section, we have described a variant of convolutional kernel networks where linear subspaces are learned at every layer. This is achieved without supervision by a K-means algorithm leading to small projection residuals. It is thus natural to introduce also a discriminative approach.

### 3.1 Backpropagation Rules for Convolutional Kernel Networks

We now consider a prediction task, where we are given a training set of images $I_0^1, I_0^2, \ldots, I_0^n$ with respective scalar labels $y_1, \ldots, y_n$ living either in $\{-1; +1\}$ for binary classification and $\mathbb{R}$ for regression. For simplicity, we only present these two settings here, but extensions to multiclass classification and multivariate regression are straightforward. We also assume that we are given a smooth convex loss function $L : \mathbb{R} \times \mathbb{R} \to \mathbb{R}$ that measures the fit of a prediction to the true label $y$.

Given a positive definite kernel $K$ on images, the classical empirical risk minimization formulation consists of finding a prediction function in the RKHS $\mathcal{H}$ associated to $K$ by minimizing the objective

$$\min_{f \in \mathcal{H}} \frac{1}{n} \sum_{i=1}^{n} L(y_i, f(I_0^i)) + \frac{\lambda}{2} \|f\|_{\mathcal{H}}^2, \tag{5}$$

where the parameter $\lambda$ controls the smoothness of the prediction function $f$ with respect to the geometry induced by the kernel, hence regularizing and reducing overfitting [22]. After training a convolutional kernel network with $k$ layers, such a positive definite kernel may be defined as

$$K_{\mathcal{Z}}(I_0, I_0') = \sum_{z \in \Omega_k} \langle f_k(z), f_k'(z) \rangle_{\mathcal{H}_k} = \sum_{z \in \Omega_k} \langle I_k(z), I_k'(z) \rangle, \tag{6}$$

where $I_k, I_k'$ are the $k$-th finite-dimensional feature maps of $I_0, I_0'$, respectively, and $f_k, f_k'$ the corresponding maps in $\Omega_k \to \mathcal{H}_k$, which have been defined in the previous section. The kernel is also indexed by $\mathcal{Z}$, which represents the network parameters—that is, the subspaces $\mathcal{F}_1, \ldots, \mathcal{F}_k$, or equivalently the set of filters $\mathbf{Z}_1, \ldots, \mathbf{Z}_k$ from Eq. (3). Then, formulation (5) becomes equivalent to

$$\min_{\mathbf{W} \in \mathbb{R}^{p_k \times |\Omega_k|}} \frac{1}{n} \sum_{i=1}^{n} L(y_i, \langle \mathbf{W}, I_k^i \rangle) + \frac{\lambda}{2} \|\mathbf{W}\|_{\mathrm{F}}^2, \tag{7}$$

where $\|.\|_{\mathrm{F}}$ is the Frobenius norm that extends the Euclidean norm to matrices, and, with an abuse of notation, the maps $I_k^i$ are seen as matrices in $\mathbb{R}^{p_k \times |\Omega_k|}$. Then, *the supervised convolutional kernel network formulation consists of jointly minimizing (7) with respect to $\mathbf{W}$ in $\mathbb{R}^{p_k \times |\Omega_k|}$ and with respect to the set of filters $\mathbf{Z}_1, \ldots, \mathbf{Z}_k$, whose columns are constrained to be on the Euclidean sphere.*

**Computing the derivative with respect to the filters $\mathbf{Z}_1, \ldots, \mathbf{Z}_k$.**
Since we consider a smooth loss function $L$, *e.g.*, logistic, squared hinge, or square loss, optimizing (7) with respect to $\mathbf{W}$ can be achieved with any gradient-based method. Moreover, when $L$ is convex, we may also use fast dedicated solvers, (see, *e.g.*, [16], and references therein). Optimizing with respect to the filters $\mathbf{Z}_j$, $j = 1, \ldots, k$ is more involved because of the lack of convexity. Yet, the objective function is differentiable, and there is hope to find a "good" stationary point by using classical stochastic optimization techniques that have been successful for training deep networks.

For that, we need to compute the gradient by using the chain rule—also called "backpropagation" [13]. We instantiate this rule in the next lemma, which we have found useful to simplify the calculation.

**Lemma 1 (Perturbation view of backpropagation.)**
*Consider an image $I_0$ represented here as a matrix in $\mathbb{R}^{p_0 \times |\Omega_0|}$, associated to a label $y$ in $\mathbb{R}$ and call $I_k^{\mathcal{Z}}$ the $k$-th feature map obtained by encoding $I_0$ with the network parameters $\mathcal{Z}$. Then, consider a perturbation $\mathcal{E} = \{\varepsilon_1, \ldots, \varepsilon_k\}$ of the set of filters $\mathcal{Z}$. Assume that we have for all $j \geq 0$,*

$$I_j^{\mathcal{Z}+\mathcal{E}} = I_j^{\mathcal{Z}} + \Delta I_j^{\mathcal{Z},\mathcal{E}} + o(\|\mathcal{E}\|), \tag{8}$$

*where $\|\mathcal{E}\|$ is equal to $\sum_{l=1}^{k} \|\varepsilon_l\|_F$, and $\Delta I_j^{\mathcal{Z},\mathcal{E}}$ is a matrix in $\mathbb{R}^{p_j \times |\Omega_j|}$ such that for all matrices $\mathbf{U}$ of the same size,*

$$\langle \Delta I_j^{\mathcal{Z},\mathcal{E}}, \mathbf{U} \rangle = \langle \varepsilon_j, g_j(\mathbf{U}) \rangle + \langle \Delta I_{j-1}^{\mathcal{Z},\mathcal{E}}, h_j(\mathbf{U}) \rangle, \tag{9}$$

*where the inner-product is the Frobenius's one and $g_j, h_j$ are linear functions. Then,*

$$\nabla_{\mathbf{Z}_j} L(y, \langle \mathbf{W}, I_k^{\mathcal{Z}} \rangle) = L'(y, \langle \mathbf{W}, I_k^{\mathcal{Z}} \rangle) \, g_j(h_{j+1}(\ldots h_k(\mathbf{W}))), \tag{10}$$

*where $L'$ denote the derivative of the smooth function $L$ with respect to its second argument.*

The proof of this lemma is straightforward and follows from the definition of the Fréchet derivative. Nevertheless, it is useful to derive the closed form of the gradient in the next proposition.

**Proposition 1 (Gradient of the loss with respect to the the filters $\mathbf{Z}_1, \ldots, \mathbf{Z}_k$.)**
*Consider the quantities introduced in Lemma 1, but denote $I_j^{\mathcal{Z}}$ by $I_j$ for simplicity. By construction, we have for all $j \geq 1$,*

$$I_j = \mathbf{A}_j \kappa_j(\mathbf{Z}_j^\top \mathbf{E}_j(I_{j-1})\mathbf{S}_j^{-1})\mathbf{S}_j \mathbf{P}_j, \tag{11}$$

*where $I_j$ is seen as a matrix in $\mathbb{R}^{p_j \times |\Omega_j|}$; $\mathbf{E}_j$ is the linear operator that extracts all overlapping $e_{j-1} \times e_{j-1}$ patches from a map such that $\mathbf{E}_j(I_{j-1})$ is a matrix of size $p_{j-1}e_{j-1}^2 \times |\Omega_{j-1}|$; $\mathbf{S}_j$ is a diagonal matrix whose diagonal entries carry the $\ell_2$-norm of the columns of $\mathbf{E}_j(I_{j-1})$; $\mathbf{A}_j$ is short for $\kappa_j(\mathbf{Z}_j^\top \mathbf{Z}_j)^{-1/2}$; and $\mathbf{P}_j$ is a matrix of size $|\Omega_{j-1}| \times |\Omega_j|$ performing the linear pooling operation. Then, the gradient of the loss with respect to the filters $\mathbf{Z}_j$, $j = 1, \ldots, k$ is given by (10) with*

$$
\begin{aligned}
g_j(\mathbf{U}) &= \mathbf{E}_j(I_{j-1})\mathbf{B}_j^\top - \frac{1}{2}\mathbf{Z}_j\big(\kappa_j'(\mathbf{Z}_j^\top \mathbf{Z}_j) \odot (\mathbf{C}_j + \mathbf{C}_j^\top)\big) \\
h_j(\mathbf{U}) &= \mathbf{E}_j^\star\big(\mathbf{Z}_j \mathbf{B}_j + \mathbf{E}_j(I_{j-1})\big(\mathbf{S}_j^{-2} \odot \big(M_j^\top \mathbf{U}\mathbf{P}_j^\top - \mathbf{E}_j(I_{j-1})^\top \mathbf{Z}_j \mathbf{B}_j\big)\big)\big),
\end{aligned} \tag{12}
$$

*where $\mathbf{U}$ is any matrix of the same size as $I_j$, $M_j = \mathbf{A}_j \kappa_j(\mathbf{Z}_j^\top \mathbf{E}_j(I_{j-1})\mathbf{S}_j^{-1})\mathbf{S}_j$ is the $j$-th feature map before the pooling step, $\odot$ is the Hadamart (elementwise) product, $\mathbf{E}_j^\star$ is the adjoint of $\mathbf{E}_j$, and*

$$\mathbf{B}_j = \kappa_j'(\mathbf{Z}_j^\top \mathbf{E}_j(I_{j-1})\mathbf{S}_j^{-1}) \odot \big(\mathbf{A}_j \mathbf{U}\mathbf{P}_j^\top\big) \quad and \quad \mathbf{C}_j = \mathbf{A}_j^{1/2} I_j \mathbf{U}^\top \mathbf{A}_j^{3/2}. \tag{13}$$

The proof is presented in Appendix B. Most quantities that appear above admit physical interpretations: multiplication by $\mathbf{P}_j$ performs downsampling; multiplication by $\mathbf{P}_j^\top$ performs upsampling; multiplication of $\mathbf{E}_j(I_{j-1})$ on the right by $\mathbf{S}_j^{-1}$ performs $\ell_2$-normalization of the columns; $\mathbf{Z}_j^\top \mathbf{E}_j(I_{j-1})$ can be seen as a spatial convolution of the map $I_{j-1}$ by the filters $\mathbf{Z}_j$; finally, $\mathbf{E}_j^\star$ "combines" a set of patches into a spatial map by adding to each pixel location the respective patch contributions.

Computing the gradient requires a forward pass to obtain the maps $I_j$ through (11) and a backward pass that composes the functions $g_j, h_j$ as in (10). The complexity of the forward step is dominated by the convolutions $\mathbf{Z}_j^\top \mathbf{E}_j(I_{j-1})$, as in convolutional neural networks. The cost of the backward pass is the same as the forward one up to a constant factor. Assuming $p_j \leq |\Omega_{j-1}|$, which is typical for lower layers that require more computation than upper ones, the most expensive cost is due to $\mathbf{E}_j(I_{j-1})\mathbf{B}_j^\top$ and $\mathbf{Z}_j \mathbf{B}_j$ which is the same as $\mathbf{Z}_j^\top \mathbf{E}_j(I_{j-1})$. We also pre-compute $\mathbf{A}_j^{1/2}$ and $\mathbf{A}_j^{3/2}$ by eigenvalue decompositions, whose cost is reasonable when performed only once per minibatch. Off-diagonal elements of $M_j^\top \mathbf{U}\mathbf{P}_j^\top - \mathbf{E}_j(I_{j-1})^\top \mathbf{Z}_j \mathbf{B}_j$ are also not computed since they are set to zero after elementwise multiplication with a diagonal matrix. In practice, we also replace $\mathbf{A}_j$ by $\big(\kappa_j(\mathbf{Z}_j^\top \mathbf{Z}_j) + \varepsilon \mathbf{I}\big)^{-1/2}$ with $\varepsilon = 0.001$, which corresponds to performing a regularized projection onto $\tilde{\mathcal{F}}_j$ (see Appendix A). Finally, a small offset of $0.00001$ is added to the diagonal entries of $\mathbf{S}_j$.

**Optimizing hyper-parameters for RBF kernels.** When using the kernel (2), the objective is differentiable with respect to the hyper-parameters $\alpha_j$. When large amounts of training data are available and overfitting is not a issue, optimizing the training loss by taking gradient steps with respect to these parameters seems appropriate instead of using a canonical parameter value. Otherwise, more involved techniques may be needed; we plan to investigate other strategies in future work.

### 3.2 Optimization and Practical Heuristics

The backpropagation rules of the previous section have set up the stage for using a stochastic gradient descent method (SGD). We now present a few strategies to accelerate it in our context.

**Hybrid convex/non-convex optimization.** Recently, many incremental optimization techniques have been proposed for solving *convex* optimization problems of the form (7) when $n$ is large but finite (see [16] and references therein). These methods usually provide a great speed-up over the stochastic gradient descent algorithm without suffering from the burden of choosing a learning rate. The price to pay is that they rely on convexity, and they require storing into memory the full training set. For solving (7) with *fixed* network parameters $\mathcal{Z}$, it means storing the $n$ maps $I_k^i$, which is often reasonable if we do not use data augmentation. To partially leverage these fast algorithms for our non-convex problem, we have adopted a minimization scheme that alternates between two steps: (i) fix $\mathcal{Z}$, then make a forward pass on the data to compute the $n$ maps $I_k^i$ and minimize the convex problem (7) with respect to $\mathbf{W}$ using the accelerated MISO algorithm [16]; (ii) fix $\mathbf{W}$, then make one pass of a projected stochastic gradient algorithm to update the $k$ set of filters $\mathbf{Z}_j$. The set of network parameters $\mathcal{Z}$ is initialized with the unsupervised learning method described in Section 2.

**Preconditioning on the sphere.** The kernels $\kappa_j$ are defined on the sphere; therefore, it is natural to constrain the filters—that is, the columns of the matrices $\mathbf{Z}_j$—to have unit $\ell_2$-norm. As a result, a classical stochastic gradient descent algorithm updates at iteration $t$ each filter $\mathbf{z}$ as follows $\mathbf{z} \leftarrow \text{Proj}_{\|.\|_2=1}[\mathbf{z} - \eta_t \nabla_{\mathbf{z}} L_t]$, where $\nabla_{\mathbf{z}} L_t$ is an estimate of the gradient computed on a minibatch and $\eta_t$ is a learning rate. In practice, we found that convergence could be accelerated by preconditioning, which consists of optimizing after a change of variable to reduce the correlation of gradient entries. For *unconstrained* optimization, this heuristic involves choosing a symmetric positive definite matrix $\mathbf{Q}$ and replacing the update direction $\nabla_{\mathbf{z}} L_t$ by $\mathbf{Q} \nabla_{\mathbf{z}} L_t$, or, equivalently, performing the change of variable $\mathbf{z} = \mathbf{Q}^{1/2} \mathbf{z}'$ and optimizing over $\mathbf{z}'$. When constraints are present, the case is not as simple since $\mathbf{Q} \nabla_{\mathbf{z}} L_t$ may not be a descent direction. Fortunately, it is possible to exploit the manifold structure of the constraint set (here, the sphere) to perform an appropriate update [1]. Concretely, (i) we choose a matrix $\mathbf{Q}$ per layer that is equal to the inverse covariance matrix of the patches from the same layer computed after the initialization of the network parameters. (ii) We perform stochastic gradient descent steps on the sphere manifold after the change of variable $\mathbf{z} = \mathbf{Q}^{1/2} \mathbf{z}'$, leading to the update $\mathbf{z} \leftarrow \text{Proj}_{\|.\|_2=1}[\mathbf{z} - \eta_t(\mathbf{I} - (1/\mathbf{z}^\top \mathbf{Q} \mathbf{z})\mathbf{Q} \mathbf{z} \mathbf{z}^\top)\mathbf{Q} \nabla_{\mathbf{z}} L_t]$. Because this heuristic is not a critical component, but simply an improvement of SGD, we relegate mathematical details in Appendix C.

**Automatic learning rate tuning.** Choosing the right learning rate in stochastic optimization is still an important issue despite the large amount of work existing on the topic, see, *e.g.*, [13] and references therein. In our paper, we use the following basic heuristic: the initial learning rate $\eta_t$ is chosen "large enough"; then, the training loss is evaluated after each update of the weights $\mathbf{W}$. When the training loss increases between two epochs, we simply divide the learning rate by two, and perform "back-tracking" by replacing the current network parameters by the previous ones.

**Active-set heuristic.** For classification tasks, "easy" samples have often negligible contribution to the gradient (see, *e.g.*, [13]). For instance, for the squared hinge loss $L(y, \hat{y}) = \max(0, 1 - y\hat{y})^2$, the gradient vanishes when the margin $y\hat{y}$ is greater than one. This motivates the following heuristic: we consider a set of active samples, initially all of them, and remove a sample from the active set as soon as we obtain zero when computing its gradient. In the subsequent optimization steps, only active samples are considered, and after each epoch, we randomly reactivate $10\%$ of the inactive ones.

## 4 Experiments

We now present experiments on image classification and super-resolution. All experiments were conducted on 8-core and 10-core 2.4GHz Intel CPUs using C++ and Matlab.

### 4.1 Image Classification on "Deep Learning" Benchmarks

We consider the datasets CIFAR-10 [12] and SVHN [19], which contain $32 \times 32$ images from $10$ classes. CIFAR-10 is medium-sized with $50\,000$ training samples and $10\,000$ test ones. SVHN is larger with $604\,388$ training examples and $26\,032$ test ones. We evaluate the performance of a 9-layer network, designed with few hyper-parameters: for each layer, we learn $512$ filters and choose the RBF kernels $\kappa_j$ defined in (2) with initial parameters $\alpha_j = 1/(0.5^2)$. Layers $1, 3, 5, 7, 9$ use $3 \times 3$ patches and a subsampling pooling factor of $\sqrt{2}$ except for layer 9 where the factor is 3; Layers $2, 4, 6, 8$ use simply $1 \times 1$ patches and no subsampling. For CIFAR-10, the parameters $\alpha_j$ are kept fixed during training, and for SVHN, they are updated in the same way as the filters. We use the squared hinge loss in a one vs all setting to perform multi-class classification (with shared filters $\mathcal{Z}$ between classes). The input of the network is pre-processed with the local whitening procedure described in [20]. We use the optimization heuristics from the previous section, notably the automatic learning rate scheme, and a gradient momentum with parameter 0.9, following [12]. The regularization parameter $\lambda$ and the number of epochs are set by first running the algorithm on a $80/20$ validation split of the training set. $\lambda$ is chosen near the canonical parameter $\lambda = 1/n$, in the range $2^i/n$, with $i = -4, \dots, 4$, and the number of epochs is at most $100$. The initial learning rate is $10$ with a minibatch size of $128$.

We present our results in Table 1 along with the performance achieved by a few recent methods without data augmentation or model voting/averaging. In this context, the best published results are obtained by the generalized pooling scheme of [14]. We achieve about $2\%$ test error on SVHN and about $10\%$ on CIFAR-10, which positions our method as a reasonably "competitive" one, in the same ballpark as the deeply supervised nets of [15] or network in network of [17].

Table 1: Test error in percents reported by a few recent publications on the CIFAR-10 and SVHN datasets without data augmentation or model voting/averaging.

| | Stoch P. [29] | MaxOut [9] | NiN [17] | DSN [15] | Gen P. [14] | SCKN (Ours) |
|---|---|---|---|---|---|---|
| CIFAR-10 | 15.13 | 11.68 | 10.41 | 9.69 | **7.62** | 10.20 |
| SVHN | 2.80 | 2.47 | 2.35 | 1.92 | **1.69** | 2.04 |

Due to lack of space, the results reported here only include a single supervised model. Preliminary experiments with no supervision show also that one may obtain competitive accuracy with wide shallow architectures. For instance, a two-layer network with (1024-16384) filters achieves 14.2% error on CIFAR-10. Note also that our unsupervised model outperforms original CKNs [18]. The best single model from [18] gives indeed 21.7%. Training the same architecture with our approach is two orders of magnitude faster and gives 19.3%. Another aspect we did not study is model complexity. Here as well, preliminary experiments are encouraging. Reducing the number of filters to 128 per layer yields indeed 11.95% error on CIFAR-10 and 2.15% on SVHN. A more precise comparison with no supervision and with various network complexities will be presented in another venue.

## 4.2   Image Super-Resolution from a Single Image

Image up-scaling is a challenging problem, where convolutional neural networks have obtained significant success [7, 8, 27]. Here, we follow [8] and replace traditional convolutional neural networks by our supervised kernel machine. Specifically, RGB images are converted to the YCbCr color space and the upscaling method is applied to the luminance channel only to make the comparison possible with previous work. Then, the problem is formulated as a multivariate regression one. We build a database of $200\,000$ patches of size $32 \times 32$ randomly extracted from the BSD500 dataset [2] after removing image 302003.jpg, which overlaps with one of the test images. $16 \times 16$ versions of the patches are build using the Matlab function imresize, and upscaled back to $32 \times 32$ by using bicubic interpolation; then, the goal is to predict high-resolution images from blurry bicubic interpolations.

The blurry estimates are processed by a 9-layer network, with $3 \times 3$ patches and 128 filters at every layer without linear pooling and zero-padding. Pixel values are predicted with a linear model applied to the 128-dimensional vectors present at every pixel location of the last layer, and we use the square loss to measure the fit. The optimization procedure and the kernels $\kappa_j$ are identical to the ones used for processing the SVHN dataset in the classification task. The pipeline also includes a pre-processing step, where we remove from input images a local mean component obtained by convolving the images with a $5 \times 5$ averaging box filter; the mean component is added back after up-scaling.

For the evaluation, we consider three datasets: Set5 and Set14 are standard for super-resolution; Kodim is the Kodak Image database, available at `http://r0k.us/graphics/kodak/`, which contains high-quality images with no compression or demoisaicing artefacts. The evaluation procedure follows [7, 8, 26, 27] by using the code from the author's web page. We present quantitative results in Table 2. For x3 upscaling, we simply used twice our model learned for x2 upscaling, followed by a 3/4 downsampling. This is clearly suboptimal since our model is not trained to up-scale by a factor 3, but this naive approach still outperforms other baselines [7, 8, 27] that are trained end-to-end. Note that [27] also proposes a data augmentation scheme at test time that slightly improves their results. In Appendix D, we also present a visual comparison between our approach and [8], whose pipeline is the closest to ours, up to the use of a supervised kernel machine instead of CNNs.

Table 2: Reconstruction accuracy for super-resolution in PSNR (the higher, the better). All CNN approaches are without data augmentation at test time. See Appendix D for the SSIM quality measure.

| Fact. | Dataset | Bicubic | SC [30] | ANR [26] | A+[26] | CNN1 [7] | CNN2 [8] | CSCN [27] | SCKN |
|---|---|---|---|---|---|---|---|---|---|
| | Set5 | 33.66 | 35.78 | 35.83 | 36.54 | 36.34 | 36.66 | 36.93 | **37.07** |
| x2 | Set14 | 30.23 | 31.80 | 31.79 | 32.28 | 32.18 | 32.45 | 32.56 | **32.76** |
| | Kodim | 30.84 | 32.19 | 32.23 | 32.71 | 32.62 | 32.80 | 32.94 | **33.21** |
| | Set5 | 30.39 | 31.90 | 31.92 | 32.58 | 32.39 | 32.75 | **33.10** | 33.08 |
| x3 | Set14 | 27.54 | 28.67 | 28.65 | 29.13 | 29.00 | 29.29 | 29.41 | **29.50** |
| | Kodim | 28.43 | 29.21 | 29.21 | 29.57 | 29.42 | 29.64 | 29.76 | **29.88** |

**Acknowledgments**

This work was supported by ANR (MACARON project ANR-14-CE23-0003-01).

## Footnotes

*Thoth team, Inria Grenoble, Laboratoire Jean Kuntzmann, CNRS, Univ. Grenoble Alpes, France.

[2]To simplify, we use zero-padding when patches are close to the image boundaries, but this is optional.

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
