[Supplementary Material · paper_supplement.pdf]

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

# A Orthogonal Projection on the Finite-Dimensional Subspace $\mathcal{F}_1$

First, we remark that the kernel $K_1$ is homogeneous such that for every patch $\mathbf{x}$ and scalar $\gamma > 0$,

$$\varphi_1(\gamma\mathbf{x}) = \gamma\varphi_1(\mathbf{x}).$$

Thus, we may assume $\mathbf{x}$ to have unit $\ell_2$-norm without loss of generality and perform the projection on $\mathcal{F}_1 = \mathrm{Span}(\varphi(\mathbf{z}_1), \dots, \varphi(\mathbf{z}_{p_1}))$ of the normalized patch, before applying the inverse rescaling.

Then, let us denote by $f_\mathbf{x}$ the orthogonal projection of a patch $\mathbf{x}$ with unit $\ell_2$-norm defined as

$$f_\mathbf{x} := \arg\min_{f \in \mathcal{F}_1} \|\varphi_1(\mathbf{x}) - f\|_{\mathcal{H}_1}^2,$$

which is equivalent to

$$f_\mathbf{x} := \sum_{j=1}^{p_1} \alpha_j^\star \varphi_1(\mathbf{z}_j) \quad \text{with} \quad \boldsymbol{\alpha}^\star \in \arg\min_{\boldsymbol{\alpha} \in \mathbb{R}^{p_1}} \left\| \varphi_1(\mathbf{x}) - \sum_{j=1}^{p_1} \alpha_j \varphi_1(\mathbf{z}_j) \right\|_{\mathcal{H}_1}^2.$$

After short calculation, we obtain

$$f_\mathbf{x} = \sum_{j=1}^{p_1} \alpha_j^\star \varphi_1(\mathbf{z}_j) \quad \text{with} \quad \boldsymbol{\alpha}^\star \in \arg\min_{\boldsymbol{\alpha} \in \mathbb{R}^{p_1}} \left[ 1 - 2\boldsymbol{\alpha}^\top \kappa_1(\mathbf{z}^\top\mathbf{x}) + \boldsymbol{\alpha}^\top \kappa_1(\mathbf{Z}^\top\mathbf{Z})\boldsymbol{\alpha} \right],$$

since the vectors $\mathbf{z}_j$ provided by the spherical K-means algorithm have unit $\ell_2$-norm. Assuming $\kappa_1(\mathbf{Z}^\top\mathbf{Z})$ to be invertible, we have $\boldsymbol{\alpha}^\star = \kappa_1(\mathbf{Z}^\top\mathbf{Z})^{-1}\kappa_1(\mathbf{Z}^\top\mathbf{x})$. After projection, normalized patches $\mathbf{x}$, $\mathbf{x}'$ may be parametrized by $\boldsymbol{\alpha}^\star = \kappa_1(\mathbf{Z}^\top\mathbf{Z})^{-1}\kappa_1(\mathbf{Z}^\top\mathbf{x})$ and $\boldsymbol{\alpha}'^\star = \kappa_1(\mathbf{Z}^\top\mathbf{Z})^{-1}\kappa_1(\mathbf{Z}^\top\mathbf{x}')$, respectively. Then, we have

$$\langle f_\mathbf{x}, f_{\mathbf{x}'} \rangle_{\mathcal{H}_1} = \boldsymbol{\alpha}^{\star\top}\kappa_1(\mathbf{Z}^\top\mathbf{Z})\boldsymbol{\alpha}'^\star = \langle \psi_1(\mathbf{x}), \psi_1(\mathbf{x}') \rangle,$$

which is the desired result.

When $\kappa_1(\mathbf{Z}^\top\mathbf{Z})$ is not invertible or simply badly conditioned, it is also common to use instead

$$\psi_1(\mathbf{x}) = \left( \kappa_1(\mathbf{Z}^\top\mathbf{Z}) + \varepsilon\mathbf{I} \right)^{-1/2}\kappa_1(\mathbf{Z}^\top\mathbf{x}),$$

where $\varepsilon > 0$ is a small regularization that improves the condition number of $\kappa_1(\mathbf{Z}^\top\mathbf{Z})$. Such a modification can be interpreted as performing a slightly regularized projection onto the finite-dimensional subspace $\mathcal{F}_1$.

# B Computation of the Gradient with Respect to the Filters

To compute the gradient of the loss function, we use Lemma 1 and start by analyzing the effect of perturbing every quantity involved in (11) such that we may obtain the desired relations (8) and (9). Before proceeding, we recall the definition of the set $\mathcal{Z} + \mathcal{E} = \{\mathbf{Z}_1 + \varepsilon_1, \dots, \mathbf{Z}_k + \varepsilon_k\}$ and the precise definition of the Landau notation $o(\|\mathcal{E}\|)$, which we use in (8). Here, it simply means a quantity that is negligible in front of the norm $\|\mathcal{E}\| = \sum_{j=1}^k \|\varepsilon_j\|_\mathrm{F}$—that is,

$$\lim_{\mathcal{E} \to 0} \frac{\left\| I_j^{\mathcal{Z}+\mathcal{E}} - I_j^{\mathcal{Z}} - \Delta I_j^{\mathcal{Z},\mathcal{E}} \right\|_\mathrm{F}}{\|\mathcal{E}\|} = 0.$$

Then, we start by initializing a recursion: $I_0^{\mathcal{Z}}$ is unaffected by the perturbation and thus $\Delta I_0^{\mathcal{Z},\mathcal{E}} = 0$. Consider now an index $j > 0$ and assume that (8) holds for $j-1$ with $\Delta I_{j-1}^{\mathcal{Z},\mathcal{E}} = O(\|\mathcal{E}\|)$.

First, we remark that

$$\mathbf{E}_j(I_{j-1}^{\mathcal{Z}+\mathcal{E}}) = \mathbf{E}_j(I_{j-1}^{\mathcal{Z}}) + \mathbf{E}_j(\Delta I_{j-1}^{\mathcal{Z},\mathcal{E}}) + o(\|\mathcal{E}\|).$$

Then, the diagonal matrix $\mathbf{S}_j$ becomes after perturbation

$$\mathbf{S}_j + \underbrace{\mathbf{S}_j^{-1} \odot \left( \mathbf{E}_j(I_{j-1}^{\mathcal{Z}})^\top \mathbf{E}_j(\Delta I_{j-1}^{\mathcal{Z},\mathcal{E}}) \right)}_{\Delta\mathbf{S}_j} + o(\|\mathcal{E}\|).$$

The inverse diagonal matrix $\mathbf{S}_j^{-1}$ becomes

$$\mathbf{S}_j^{-1} \underbrace{-\mathbf{S}_j^{-3} \odot \left(\mathbf{E}_j(I_{j-1}^{\mathcal{Z}})^\top \mathbf{E}_j(\Delta I_{j-1}^{\mathcal{Z},\mathcal{E}})\right)}_{\Delta \mathbf{S}_j^{-1}} + o(\|\mathcal{E}\|),$$

and the matrix $\mathbf{A}_j$ becomes

$$
\begin{aligned}
\kappa_j\big((\mathbf{Z}_j+\varepsilon_j)^\top(\mathbf{Z}_j+\varepsilon_j)\big)^{-\frac{1}{2}} &= \kappa_j\big(\mathbf{Z}_j^\top\mathbf{Z}_j+\varepsilon_j^\top\mathbf{Z}_j+\mathbf{Z}_j^\top\varepsilon_j + o(\|\varepsilon_j\|_{\mathrm{F}})\big)^{-\frac{1}{2}} \\
&= \big(\kappa_j(\mathbf{Z}_j^\top\mathbf{Z}_j)+\kappa_j'(\mathbf{Z}_j^\top\mathbf{Z}_j)\odot(\mathbf{Z}_j^\top\varepsilon_j+\varepsilon_j^\top\mathbf{Z}_j)+o(\|\mathcal{E}\|)\big)^{-\frac{1}{2}} \\
&= \mathbf{A}_j^{\frac{1}{2}}\big(\mathbf{I}+\mathbf{A}_j\big(\kappa_j'(\mathbf{Z}_j^\top\mathbf{Z}_j)\odot(\mathbf{Z}_j^\top\varepsilon_j+\varepsilon_j^\top\mathbf{Z}_j)\big)\mathbf{A}_j+o(\|\mathcal{E}\|)\big)^{-1/2}\mathbf{A}_j^{\frac{1}{2}} \\
&= \mathbf{A}_j \underbrace{-\frac{1}{2}\mathbf{A}_j^{\frac{3}{2}}\big(\kappa_j'(\mathbf{Z}_j^\top\mathbf{Z}_j)\odot(\mathbf{Z}_j^\top\varepsilon_j+\varepsilon_j^\top\mathbf{Z}_j)\big)\mathbf{A}_j^{\frac{3}{2}}}_{\Delta\mathbf{A}_j} + o(\|\mathcal{E}\|),
\end{aligned}
$$

where we have used the relation $(\mathbf{I}+\mathbf{Q})^{-1/2} = \mathbf{I} - \frac{1}{2}\mathbf{Q} + o(\|\mathbf{Q}\|_{\mathrm{F}})$. Note that the quantities $\Delta\mathbf{A}_j, \Delta\mathbf{S}_j, \Delta\mathbf{S}_j^{-1}$ that we have introduced are all $O(\|\mathcal{E}\|)$. Then, by replacing the quantities $\mathbf{A}_j, \mathbf{S}_j, \mathbf{S}_j^{-1}, \mathbf{I}_{j-1}$ by their perturbed versions in the definition of $I_j$ given in (11), we obtain that $I_j^{\mathcal{Z}+\mathcal{E}}$ is equal to

$$(\mathbf{A}_j+\Delta\mathbf{A}_j)\kappa_j\Big((\mathbf{Z}_j+\varepsilon_j)^\top\Big(\mathbf{E}_j(I_{j-1}^{\mathcal{Z}})+\mathbf{E}_j(\Delta I_{j-1}^{\mathcal{Z},\mathcal{E}})\Big)(\mathbf{S}_j^{-1}+\Delta\mathbf{S}_j^{-1})\Big)(\mathbf{S}_j+\Delta\mathbf{S}_j)\mathbf{P}_j + o(\|\mathcal{E}\|).$$

Then, after short calculation, we obtain the desired relation $I_j^{\mathcal{Z}+\mathcal{E}} = I_j^{\mathcal{Z}+\mathcal{E}} + \Delta I_j^{\mathcal{Z},\mathcal{E}} + o(\|\mathcal{E}\|)$ with

$$
\begin{aligned}
\Delta I_j^{\mathcal{Z},\mathcal{E}} =\ & \Delta\mathbf{A}_j\kappa_j(\mathbf{Z}_j^\top\mathbf{E}_j(I_{j-1}^{\mathcal{Z}})\mathbf{S}_j^{-1})\mathbf{S}_j\mathbf{P}_j \\
& + \mathbf{A}_j\big(\kappa_j'(\mathbf{Z}_j^\top\mathbf{E}_j(I_{j-1}^{\mathcal{Z}})\mathbf{S}_j^{-1})\odot(\varepsilon_j^\top\mathbf{E}_j(I_{j-1}^{\mathcal{Z}}))\big)\mathbf{P}_j \\
& + \mathbf{A}_j\Big(\kappa_j'(\mathbf{Z}_j^\top\mathbf{E}_j(I_{j-1}^{\mathcal{Z}})\mathbf{S}_j^{-1})\odot(\mathbf{Z}_j^\top\mathbf{E}_j(\Delta I_{j-1}^{\mathcal{Z},\mathcal{E}}))\Big)\mathbf{P}_j \\
& + \mathbf{A}_j\big(\kappa_j'(\mathbf{Z}_j^\top\mathbf{E}_j(I_{j-1}^{\mathcal{Z}})\mathbf{S}_j^{-1})\odot(\mathbf{Z}_j^\top\mathbf{E}_j(I_{j-1}^{\mathcal{Z}})\Delta\mathbf{S}_j^{-1}\mathbf{S}_j))\big)\mathbf{P}_j \\
& + \mathbf{A}_j\kappa_j(\mathbf{Z}_j^\top\mathbf{E}_j(I_{j-1}^{\mathcal{Z}})\mathbf{S}_j^{-1})\Delta\mathbf{S}_j\mathbf{P}_j.
\end{aligned}
$$

First, we remark that $\Delta I_j^{\mathcal{Z},\mathcal{E}} = O(\|\mathcal{E}\|)$, which is one of the induction hypothesis we need. Then, after plugging in the values of $\Delta\mathbf{A}_j, \Delta\mathbf{S}_j, \Delta\mathbf{S}_j^{-1}$, and with further simplification, we obtain

$$
\begin{aligned}
\Delta I_j^{\mathcal{Z},\mathcal{E}} =\ & -\frac{1}{2}\mathbf{A}_j^{\frac{3}{2}}\big(\kappa_j'(\mathbf{Z}_j^\top\mathbf{Z}_j)\odot(\mathbf{Z}_j^\top\varepsilon_j+\varepsilon_j^\top\mathbf{Z}_j)\big)\mathbf{A}_j^{\frac{1}{2}}I_j^{\mathcal{Z}} \\
& + \mathbf{A}_j\big(\kappa_j'(\mathbf{Z}_j^\top\mathbf{E}_j(I_{j-1}^{\mathcal{Z}})\mathbf{S}_j^{-1})\odot(\varepsilon_j^\top\mathbf{E}_j(I_{j-1}^{\mathcal{Z}}))\big)\mathbf{P}_j \\
& + \mathbf{A}_j\Big(\kappa_j'(\mathbf{Z}_j^\top\mathbf{E}_j(I_{j-1}^{\mathcal{Z}})\mathbf{S}_j^{-1})\odot\Big(\mathbf{Z}_j^\top\mathbf{E}_j(\Delta I_{j-1}^{\mathcal{Z},\mathcal{E}})\Big)\Big)\mathbf{P}_j \\
& - \mathbf{A}_j\Big(\kappa_j'(\mathbf{Z}_j^\top\mathbf{E}_j(I_{j-1}^{\mathcal{Z}})\mathbf{S}_j^{-1})\odot\Big(\mathbf{Z}_j^\top\mathbf{E}_j(I_{j-1}^{\mathcal{Z}})\Big(\mathbf{S}_j^{-2}\odot\Big(\mathbf{E}_j(I_{j-1}^{\mathcal{Z}})^\top\mathbf{E}_j(\Delta I_{j-1}^{\mathcal{Z},\mathcal{E}})\Big)\Big)\Big)\Big)\mathbf{P}_j \\
& + M_j^{\mathcal{Z}}\Big(\mathbf{S}_j^{-2}\odot\Big(\mathbf{E}_j(I_{j-1}^{\mathcal{Z}})^\top\mathbf{E}_j(\Delta I_{j-1}^{\mathcal{Z},\mathcal{E}})\Big)\Big)\mathbf{P}_j,
\end{aligned}
$$

where $M_j^{\mathcal{Z}}$ is the $j$-th feature map of $I_0$ before the $j$-th linear pooling step—that is, $I_j^{\mathcal{Z}} = M_j^{\mathcal{Z}}\mathbf{P}_j$. We now see that $\Delta I_j^{\mathcal{Z},\mathcal{E}}$ is linear in $\varepsilon_j$ and $\Delta I_{j-1}^{\mathcal{Z},\mathcal{E}}$, which guarantees that there exist two linear functions $g_j, h_j$ that satisfy (9). More precisely, we want for all matrix $\mathbf{U}$ of the same size as $\Delta I_j^{\mathcal{Z},\mathcal{E}}$

$$
\begin{aligned}
\langle\varepsilon_j, g_j(\mathbf{U})\rangle = \ & \Big\langle -\frac{1}{2}\mathbf{A}_j^{\frac{3}{2}}\big(\kappa_j'(\mathbf{Z}_j^\top\mathbf{Z}_j)\odot(\mathbf{Z}_j^\top\varepsilon_j+\varepsilon_j^\top\mathbf{Z}_j)\big)\mathbf{A}_j^{\frac{1}{2}}I_j^{\mathcal{Z}}, \mathbf{U}\Big\rangle \\
& + \big\langle\mathbf{A}_j\big(\kappa_j'(\mathbf{Z}_j^\top\mathbf{E}_j(I_{j-1}^{\mathcal{Z}})\mathbf{S}_j^{-1})\odot(\varepsilon_j^\top\mathbf{E}_j(I_{j-1}^{\mathcal{Z}}))\big)\mathbf{P}_j, \mathbf{U}\big\rangle,
\end{aligned}
$$

and

$$\langle \Delta I_{j-1}^{\mathcal{Z},\mathcal{E}}, h_j(\mathbf{U})\rangle = \Big\langle \mathbf{A}_j\Big(\kappa_j'(\mathbf{Z}_j^\top \mathbf{E}_j(I_{j-1}^{\mathcal{Z}})\mathbf{S}_j^{-1}) \odot \big(\mathbf{Z}_j^\top \mathbf{E}_j(\Delta I_{j-1}^{\mathcal{Z},\mathcal{E}})\big)\Big)\mathbf{P}_j, \mathbf{U}\Big\rangle$$
$$- \Big\langle \mathbf{A}_j\Big(\kappa_j'(\mathbf{Z}_j^\top \mathbf{E}_j(I_{j-1}^{\mathcal{Z}})\mathbf{S}_j^{-1}) \odot \big(\mathbf{Z}_j^\top \mathbf{E}_j(I_{j-1}^{\mathcal{Z}})\big(\mathbf{S}_j^{-2} \odot \big(\mathbf{E}_j(I_{j-1}^{\mathcal{Z}})^\top \mathbf{E}_j(\Delta I_{j-1}^{\mathcal{Z},\mathcal{E}})\big)\big)\big)\Big)\mathbf{P}_j, \mathbf{U}\Big\rangle$$
$$+ \Big\langle M_j^{\mathcal{Z}}\Big(\mathbf{S}_j^{-2} \odot \big(\mathbf{E}_j(I_{j-1}^{\mathcal{Z}})^\top \mathbf{E}_j(\Delta I_{j-1}^{\mathcal{Z},\mathcal{E}})\big)\Big)\mathbf{P}_j, \mathbf{U}\Big\rangle.$$

Then, it is easy to obtain the form of $g_j, h_j$ given in (12), by using in the right order the following elementary calculus rules: (i) $\langle \mathbf{UV}, \mathbf{W}\rangle = \langle \mathbf{U}, \mathbf{WV}^\top\rangle = \langle \mathbf{V}, \mathbf{U}^\top \mathbf{W}\rangle$, (ii) $\langle \mathbf{U}, \mathbf{V}\rangle = \langle \mathbf{U}^\top, \mathbf{V}^\top\rangle$, (iii) $\langle \mathbf{U} \odot \mathbf{V}, \mathbf{W}\rangle = \langle \mathbf{U}, \mathbf{V} \odot \mathbf{W}\rangle$ for any matrices $\mathbf{U}, \mathbf{V}, \mathbf{W}$ of appropriate sizes, and also (iv) $\langle \mathbf{E}_j(\mathbf{U}), \mathbf{V}\rangle = \langle \mathbf{U}, \mathbf{E}_j^\star(\mathbf{V})\rangle$, by definition of the adjoint operator. We conclude by induction.

## C   Preconditioning Heuristic on the Sphere

In this section, we present a preconditioning heuristic for optimizing over the sphere $\mathbb{S}^{p-1}$, inspired by second-order (Newton) optimization techniques on smooth manifolds [1]. Following [1], we will consider gradient descent steps on the manifold. A fundamental operation is thus the projection operator $P_{\mathbf{z}}$ onto the tangent space at a point $\mathbf{z}$. This operator is defined for the sphere by

$$P_{\mathbf{z}}[\mathbf{u}] = (\mathbf{I} - \mathbf{z}\mathbf{z}^\top)\mathbf{u},$$

for any vector $\mathbf{u}$ in $\mathbb{R}^p$. Another important operator is the Euclidean projection on $\mathbb{S}^{p-1}$, which was denoted by $\mathrm{Proj}_{\|.\|_2=1}$ in previous parts of the paper.

**Gradient descent on the sphere $\mathbb{S}^{p-1}$ is equivalent to the projected gradient descent in $\mathbb{R}^p$.**
When optimizing on a manifold, the natural descent direction is the projected gradient $P_{\mathbf{z}}\nabla L(\mathbf{z})$. In the case of the sphere, a gradient step on the manifold is equivalent to a classical projected gradient descent step in $\mathbb{R}^p$ with particular step size:

$$\mathrm{Proj}_{\|.\|_2=1}[\mathbf{z} - \eta P_{\mathbf{z}}[\nabla L(\mathbf{z})]] = \mathrm{Proj}_{\|.\|_2=1}\big[\mathbf{z} - \eta(\mathbf{I} - \mathbf{z}\mathbf{z}^\top)\nabla L(\mathbf{z})\big]$$
$$= \mathrm{Proj}_{\|.\|_2=1}\big[\big(1 + \eta\mathbf{z}^\top \nabla L(\mathbf{z})\big)\mathbf{z} - \eta\nabla L(\mathbf{z})\big]$$
$$= \mathrm{Proj}_{\|.\|_2=1}\Big[\mathbf{z} - \frac{\eta}{1 + \eta\mathbf{z}^\top \nabla L(\mathbf{z})}\nabla L(\mathbf{z})\Big].$$

**In $\mathbb{R}^p$ with no constraint, pre-conditioning is equivalent to performing a change of variable.**
For *unconstrained* optimization in $\mathbb{R}^p$, faster convergence is usually achieved when one has access to an estimate of the inverse of the Hessian $\nabla^2 L(\mathbf{z})$—assuming twice differentiability—and using the descent direction $(\nabla^2 L(\mathbf{z}))^{-1}\nabla L(\mathbf{z})$ instead of $\nabla L(\mathbf{z})$; then, we obtain a Newton method. When the exact Hessian is not available, or too costly to compute and/or invert, it is however common to use instead a constant estimate of the inverse Hessian, denoted here by $\mathbf{Q}$, which we call pre-conditioning matrix. Finding an appropriate matrix $\mathbf{Q}$ is difficult in general, but for learning linear models, a typical choice is to use the inverse covariance matrix of the data (or one approximation). In that case, the preconditioned gradient descent step consists of the update $\mathbf{z} - \eta\mathbf{Q}\nabla L(\mathbf{z})$. Such a matrix $\mathbf{Q}$ is defined similarly in the context of convolutional kernel networks, as explained in the main part of the paper. A useful interpretation of preconditioning is to see it as optimizing after a change of variable. Define indeed the objective

$$\tilde{L}(\mathbf{w}) = L(\mathbf{Q}^{1/2}\mathbf{w}).$$

Then, minimizing $\tilde{L}$ is equivalent to minimizing $L$ with respect to $\mathbf{z}$, with the relation $\mathbf{z} = \mathbf{Q}^{1/2}\mathbf{w}$. Moreover, when there is no constraint on $\mathbf{z}$ and $\mathbf{w}$, the regular gradient descent algorithm on $\tilde{L}$ is equivalent to the preconditioned gradient descent on $L$:

$$\mathbf{w} \leftarrow \mathbf{w} - \eta\nabla \tilde{L}(\mathbf{w}) \iff \mathbf{w} \leftarrow \mathbf{w} - \eta\mathbf{Q}^{1/2}\nabla L(\mathbf{Q}^{1/2}\mathbf{w})$$
$$\iff \mathbf{z} \leftarrow \mathbf{z} - \eta\mathbf{Q}\nabla L(\mathbf{z}) \quad \text{with} \quad \mathbf{z} = \mathbf{Q}^{1/2}\mathbf{w}.$$

We remark that the Hessian $\nabla^2 \tilde{L}(\mathbf{w})$ is equal to $\mathbf{Q}^{1/2}\nabla^2 L(\mathbf{Q}^{1/2}\mathbf{w})\mathbf{Q}^{1/2}$, which is equal to identity when $\mathbf{Q}$ coincides with the inverse Hessian of $L$. In general, this is of course not the case, but the hope is to obtain a Hessian $\nabla^2 \tilde{L}$ that is better conditioned than $\nabla^2 L$, thus resulting in faster convergence.

**Preconditioning on a smooth manifold requires some care.**

Unfortunately, using second-order information (or simply a pre-conditioning matrix) when optimizing over a constraint set or over a smooth manifold is not as simple as optimizing in $\mathbb{R}^p$ since the quantities $\mathbf{Q}\nabla L(\mathbf{z}), P_z[\mathbf{Q}\nabla L(\mathbf{z})], \mathbf{Q}P_z[\nabla L(\mathbf{z})]$ may not be feasible descent directions. However, the point of view that sees pre-conditioning as a change of variable will give us the right direction to follow.

Optimizing $L$ on $\mathbb{S}^{p-1}$ is in fact equivalent to optimizing $\tilde{L}$ on the smooth manifold

$$\tilde{\mathbb{S}}^{p-1} = \left\{ \mathbf{w} \in \mathbb{R}^p : \|\mathbf{Q}^{1/2}\mathbf{w}\|_2 = 1 \right\},$$

which represents an ellipsoid. The tangent plane at a point $\mathbf{w}$ of $\tilde{\mathbb{S}}^{p-1}$ being defined by the normal vector $\mathbf{Q}\mathbf{w}/\|\mathbf{Q}\mathbf{w}\|_2$, it is then possible to introduce the projection operator $\tilde{P}_{\mathbf{w}}$ on the tangent space:

$$\tilde{P}_{\mathbf{w}}[\mathbf{u}] = \left( \mathbf{I} - \frac{\mathbf{Q}\mathbf{w}\mathbf{w}^\top\mathbf{Q}}{\mathbf{w}^\top\mathbf{Q}^2\mathbf{w}} \right)\mathbf{u}.$$

Then, we may define the gradient descent step rule on $\tilde{\mathbb{S}}^{p-1}$ as

$$\mathbf{w} \leftarrow \mathrm{Proj}_{\tilde{\mathbb{S}}^{p-1}}\left[ \mathbf{w} - \eta\tilde{P}_{\mathbf{w}}\left[ \nabla\tilde{L}(\mathbf{w}) \right] \right] = \mathrm{Proj}_{\tilde{\mathbb{S}}^{p-1}}\left[ \mathbf{w} - \eta\left( \mathbf{I} - \frac{\mathbf{Q}\mathbf{w}\mathbf{w}^\top\mathbf{Q}}{\mathbf{w}^\top\mathbf{Q}^2\mathbf{w}} \right)\mathbf{Q}^{1/2}\nabla L(\mathbf{Q}^{1/2}\mathbf{w}) \right].$$

With the change of variable $\mathbf{z} = \mathbf{Q}^{1/2}\mathbf{w}$, this is equivalent to

$$\mathbf{z} \leftarrow \mathrm{Proj}_{\|\cdot\|_2 = 1}\left[ \mathbf{z} - \eta\left( \mathbf{I} - \frac{\mathbf{Q}\mathbf{z}\mathbf{z}^\top}{\mathbf{z}^\top\mathbf{Q}\mathbf{z}} \right)\mathbf{Q}\nabla L(\mathbf{z}) \right].$$

This is exactly the update rule we have chosen in our paper, as a heuristic in a stochastic setting.

# D    Additional Results for Image Super-Resolution

We present a quantitative comparison in Table 3 using the structural similarity index measure (SSIM), which is known to better reflect the quality perceived by humans than the PSNR; it is commonly used to evaluate the quality of super-resolution methods, see [8, 26, 27]. Then, we present a visual comparison between several approaches in Figures 2, 3, and 4. We focus notably on the classical convolutional neural network of [8] since our pipeline essentially differs in the use of our supervised kernel machine instead of convolutional neural networks. After subjective evaluation, we observe that both methods perform equally well in textured areas. However, our approach recovers better thin high-frequency details, such as the eyelash of the baby in the first image. By zooming on various parts, it is easy to notice similar differences in other images. We also observed a few ghosting artefacts near object boundaries with the method of [8], which is not the case with our approach.

Table 3: Reconstruction accuracy of various super-resolution approaches. The numbers represent the structural similarity index (SSIM), the higher, the better.

| Fact. | Dataset | Bicubic | SC [30] | ANR [26] | A+[26] | CNN1 [7] | CNN2 [8] | CSCN [27] | SCKN |
|-------|---------|---------|---------|----------|--------|----------|----------|-----------|--------|
|       | Set5    | 0.9299  | 0.9492  | 0.9499   | 0.9544 | 0.9521   | 0.9542   | 0.9552    | **0.9580** |
| x2    | Set14   | 0.8689  | 0.8989  | 0.9004   | 0.9056 | 0.9037   | 0.9067   | 0.9074    | **0.9115** |
|       | Kodim   | 0.8684  | 0.8990  | 0.9007   | 0.9075 | 0.9043   | 0.9068   | 0.9104    | **0.9146** |
|       | Set5    | 0.8677  | 0.8959  | 0.8959   | 0.9088 | 0.9025   | 0.9090   | 0.9144    | **0.9165** |
| x3    | Set14   | 0.7741  | 0.8074  | 0.8092   | 0.8188 | 0.8148   | 0.8215   | 0.8238    | **0.8297** |
|       | Kodim   | 0.7768  | 0.8066  | 0.8084   | 0.8175 | 0.8109   | 0.8174   | 0.8222    | **0.8283** |

| Bicubic | Sparse coding [30] | CNN2 [8] | SCKN (Ours) |

Figure 2: Visual comparison for x3 image up-scaling. Each column corresponds to a different method (see bottom row). RGB images are converted to the YCbCr color space and the up-scaling method is applied to the luminance channel only. Color channels are up-scaled using bicubic interpolation for visualization purposes. CNN2 and SCKN perform similarly in textured areas, but SCKN provides significantly sharper artefact-free edges (see in particular the butterfly image). Best seen by zooming on a computer screen with an appropriate PDF viewer that does not smooth the image content.

| Bicubic | Sparse coding [30] | CNN2 [8] | SCKN (Ours) |

Figure 3: Another visual comparison for x3 image up-scaling. See caption of Figure 2. Best seen by zooming on a computer screen with an appropriate PDF viewer that does not smooth the image content.

|              |                    |           |             |
|:------------:|:------------------:|:---------:|:-----------:|
| Bicubic      | Sparse coding [30] | CNN2 [8]  | SCKN (Ours) |

Figure 4: Another visual comparison for x3 image up-scaling. See caption of Figure 2. Best seen by zooming on a computer screen with an appropriate PDF viewer that does not smooth the image content.