[Reviews · NeurIPS 2016]

Reviewer 1

Summary

This work proposes an supervised enhancement to convolutional kernel networks (CKN) for image classification and super-resolution. The main contribution is a scheme to project data onto a subspace of the RKHS spanned by k-means centroids similar to RBF networks. Furthermore, these centroids may be refined in a supervised setting by applying back-propagation. In CKNs, the feature map of the predefined image patch kernel is approximated. The authors point out, as a drawback of the original method, that those approximations do not necessarily live in the RKHS. An explanation, why this is a drawback, i.e., just conceptionally or also in terms of performance/training/... is missing. In the supervised setting, the k-means centroids Z are only used as an initialization and trained using back-propagation. The paper would benefit from an evaluation how this procedure helps compared to randomly initialized Z. Is it guaranteed that Z stays within the RKHS or is the "drawback" of the original method introduced again? The preconditioning of Z on the sphere is explained by the choice of the kernel of image patches. However, the choice of this kernel is not motivated. Again, an empirically evalutation would help here. While the image classification results are comparable at best, the proposed method achieves state of the art results in the super-resolution setting. Here, error bars and a visual comparison to CNN3, which performs second-best in most settings, would have been nice.

Qualitative Assessment

In general, the paper would benefit from a clearer structure. Especially section 2 is hard to read since it mixes a rough description of the original CKN model with generalizations and enhancements suggested in this work. The motivation of merging ideas from deep learning and kernel learning is interesting, however, seems to get a bit lost. The paper would benefit from highlighting more clearly how concepts from kernel learning help in a multiplayered, feature-learning approach.

Confidence in this Review

3-Expert (read the paper in detail, know the area, quite certain of my opinion)


Reviewer 2

Summary

This paper proposes convolutional kernel networks (CKNs) for supervised learning, which is inspired by the idea of unsupervised CKNs proposed in NIPS2014. Unlike the unsupervised one, the proposed model is modeled so as to predict target variables directly as output of CKNs. Then, this paper describes a optimization procedure for its own parameters based on back-propagation. In experiments, this paper shows that the proposed model is competitive to recent CNN-based methods in image classification task and outperforms the others in image super-resolution tasks from a single image.

Qualitative Assessment

This paper is well written, and the motivation of the research is clear. Since the proposed model is formalized as a general supervised learning framework for images, it is expected that the model is applied to various applications. The experimental results show that the proposed model is promising. With the novelty, it is difficult to understand the difference between the formulations of unsupervised CKN [18] and the proposed model except the difference of their objective functions. It is desirable to make the difference more clear.

Confidence in this Review

2-Confident (read it all; understood it all reasonably well)


Reviewer 3

Summary

This paper follows the same framework of Mairal et al. [18], presented in NIPS 2014. This submitted paper has several novelties that make the new contributions relevant to be presented in NIPS 2016.

Qualitative Assessment

First of all, it is not clear why this work is entitled "end-to-end", and we find it inappropriate for this work. The motivations given in the paper are not convincing. We think that the paper can be largely improved if the author provides the list of new contributions compared to [18]. The method given in [18] should also be included in the comparative experiments. The presentation of this paper is difficult to follow. For instance, it is not clear how the two optimization problems given in Section 3.1 are jointly solved. Moreover, it is difficult to understand how (7) is equivalent to (5) since the matrix W is not defined in the paper. Some typos, such as "Hadamart"

Confidence in this Review

2-Confident (read it all; understood it all reasonably well)


Reviewer 4

Summary

This paper provides a supervised manner to leran a multilayer kernel machine, which is an extension of convolutional kernel networks (without supervision). Its major purpose is to couple learning and data representation together within the context of kernel machines.

Qualitative Assessment

This paper proposes an original idea and theoretically appealing solutions to solve it. Quality: Its first part (Section 1, 2) is excellent, but its latter part may be a little weak. Section 3 is a little dense with numerous details and heuristics. A pseudo-code showing the overall framework may be helpful for readers. Section 4 is a little short to validate the potential effectiveness of the proposed method. A direct comparison between CKN and CNN with same architecture may be more informative for readers. Originality: The purpose of this paper is appealing and its solutions are novel. It is worth an effort to explore the bridge between kernel machines and deep neural networks. But experimental results may not be enough to express its effectiveness fairly. Several direct comparisons will make this paper a more influential one. Potential impact: Its motivation is important, but more well-designed experiments will definitely make the effectiveness of this paper easier to accept. Clarity: I think Section 3 a little dense, a pseudo-code summarizing details will be welcome. Detailed Comments: (1) In the context of supervised learning, this paper seems to put less efforts to explain why the proposed solution would be superior to a standard CNN. I am not sure its natural regularization function (the norm in the RKHS) will be the key to make itself a competitive alternative to CNN. (2) Since the superiority may not be easily evaluated in an analytical form, it will be more informative to validate them empirically. Otherwise, the discussion may, by chance, mislead readers falling into the assumption that a CKN is doomed to be better than a CNN. (3) The discussion in the image classification experiments (Section 4.1) may not be sufficient. For example, why a 9-layer network is chosen? Will a shallower CKN be more suitable for CIFAR-10? Moreover, it will be more informative to provide the performance of CNNs with same architecture. It can be understood if dropout and batch-normalization are removed in the training course of CNN, but it is important to show the superiority of supervised CKN in the same architecture. Also, it will be beneficial to show the performance gain compared to its unsupervised counterpart (learnt by a k-means algorithm in all layers). Besides, the k-means algorithm is sensitive to initialization, whether the supervised CKN will still achieve competitive performance when the result of k-means is not good. Repeating the experiment several times or tuning the network from a random initialization will give more insights to readers. At last, I have several personal curiosities about the method. a) How many patches is required for the k-means algorithm to output a good initialization? b) Whether transferring weights from a pre-trained network will boost its performance. c) Whether fine-tuning the 2-layer CKN-GM or CKN-PM in [18] will show improvements. I personally think this is the key to show the effectiveness of the proposed solution. (4) The image super-resolution experiment may have the same problem. Why is a 9-layer network preferred? Since [8] claimed that a deeper network is hard to output a better result compared to a shallower one, it will be beneficial to discuss the reason that the supervised CKN can tackle this problem. I also wonder whether the performance of a 3-layer CKN will outperform its counterpart CNN in [8]. Moreover, in the context of super-resolution, inference time is also important to evaluate a method. I think the 9-layer CKN with 128 filters for each layer may be too computationally heavy for this task. It is helpful to provide a visualization of the trade-off between accuracy and inference time for different methods like Figure 1 in [26]. Summary: Overall, it is a promising paper with potentially huge influence. Due to the limited space, a tradeoff between details and experiments will surely make itself a shining one in the NIPS.

Confidence in this Review

2-Confident (read it all; understood it all reasonably well)


Reviewer 5

Summary

This paper proposed a multilayer kernel machine that performs end-to-end learning for image representation. Such a heirarchical model generalized convolutional kernel networks, and learns the kernel for a supervised prediction problem. Backpropagation rules were derived to optimize model parameters, with closed-forms obtained. It turned out that at each layer, learning filters is equivalent to optimizing a linear subspace in RKHS where data is projected. The norm in the RKHS comes as natural regularization. Competitive performance are shown on image classification and super-resolution task.

Qualitative Assessment

In short, this paper is very novel and solid. The reviewer is impressed how the kernel composition is related to (and nicely solved by) backpropagation rules. The fact that learning filters is equivalent to optimizing a linear subspace in RKHS is also inspiring. The reviewer is curious whether enforcing a kernel structure would make deep modele more "effective", e.g., whether the same representation power could be learnt with less data or parameter (as some information has been embedded as a prior when designing the specific archtiecture). In Tables 1 and 2, the performance of comparision methods are obtained from orginal papers/codes. As understood by the reviewers, the model parameter complexities (#parameters) were not mannually controlled to be identical during experiments. It thus remains skeptical whether the better performance is derived from the specific kernel strcuture, or a larger number of parameters. While completely understanding that this's (probably) a common practice in deep learning papers, the reviewer hopes the author could compare and address parameter complexities of methods, to complement this very good work.

Confidence in this Review

3-Expert (read the paper in detail, know the area, quite certain of my opinion)


Reviewer 6

Summary

The authors propose a new model of convolutional kernel network that enables end-to-end supervised training. Specifically, they first map the local image patches to infinite-dimensional vectors in a RKHS, then project the vectors onto a finite-dimensional subspace so that the coordinates of the projections can be used as easier representations. After the projection the authors use linear pooling to ensure invariance. A deep model can be constructed by stacking several such layers. Note that the bases of the projection subspace can be learned in a unsupervised way by spherical K-means or end-to-end supervised way via backpropagation. Experiments on image classification show competitive results. The authors also obtain the state-of-the-art results on several image super-resolution tasks.

Qualitative Assessment

As far as I am concerned, the work has several strengths: - bridge the gap between kernel methods and deep models and show that the performance of deep kernel models can match conventional deep neural networks - the idea of finite-dimensional subspace projection is inspiring - the framework enables end-to-end learning, including kernel parameters - the competitive results on image classification and image super-resolution (new state-of-the-art) may arouse considerable interest in the field of computer vision The main concern is lack of a GPU implementation and parallel structures of CNN such as dropout, batch normalization and so on, which the authors leave for future investigation

Confidence in this Review

2-Confident (read it all; understood it all reasonably well)